# Population Parameters and Growth of *Riptortus pedestris* (Fabricius) (Hemiptera: Alydidae) under Fluctuating Temperature

**DOI:** 10.3390/insects13020113

**Published:** 2022-01-21

**Authors:** Jeong Joon Ahn, Kyung San Choi

**Affiliations:** Research Institute of Climate Change and Agriculture, National Institute of Horticultural and Herbal Science, RDA, 1285 Aejo-roo, Jeju-do, Jeju-si 63240, Korea; mutant8@korea.kr

**Keywords:** *Riptortus pedestris*, life table, population projection, temperature

## Abstract

**Simple Summary:**

The bean bug, *Riptortus pedestris,* is a polyphagous species that is an important pest of soybean fields in Asian countries. In this study, we examined the effects of constant and fluctuating temperatures on the development and reproduction of *R. pedestris*. The effects of thermal conditions were assessed by constructing age-stage, two-sex life tables from a constant temperature of 24 °C and simulated fluctuating temperatures of 24 ± 4 °C, 24 ± 6 °C, and 24 ± 8 °C. At a constant temperature, a number of *R. pedestris* life table parameters differed from those under fluctuating temperatures. Although similar pre-adult development periods were recorded under constant and fluctuating temperatures, the periods of female longevity and oviposition periods under fluctuating temperatures were significantly longer than those at a constant temperature. Given that temperature is an important abiotic factor for forecasting the population dynamics of arthropods in nature, determining the effects of fluctuating temperatures will make a valuable contribution to predicting *R. pedestris* population growth.

**Abstract:**

In this study, we determined the influence of fluctuating temperatures on the development and fecundity of the bean bug *Riptortus pedestris* (Fabricius) (Hemiptera: Alydidae) by collecting life table data for individuals exposed at a constant temperature (24 °C) and three fluctuating temperatures (24 ± 4 °C, 24 ± 6 °C, and 24 ± 8 °C). The raw life history data were analyzed using an age-stage, two-sex life table to take into account the viable development rate among individuals. Based on these analyses, the population projections enabled us to determine the stage structure and variability of population growth under different temperature treatments. Our results revealed shorter periods of immature development and a higher pre-adult survival rate at 24 ± 6 °C than under the other assessed temperature conditions. Furthermore, significant reductions in female longevity were recorded at 24 °C, whereas the fecundity, net reproductive rate, and intrinsic and finite rates of increase were highest at 24 ± 6 °C. These findings reveal that fluctuating temperatures have a positive influence on the life history traits of *R. pedestris* and indicate that observations made under constant temperatures may not explain sufficiently enough the temperature dependent biological performances of pests in the field.

## 1. Introduction

The bean bug *Riptortus pedestris* (Fabricius) (Hemiptera: Alydidae) is an important pest of soybean fields in Asian countries, including China, India, Japan, and South Korea [1,2,3,4,5], and is the major insect pest responsible for staygreen syndrome in Chinese soybean [5]. *R. pedestris* attack the soybean pod and seeds by piercing and sucking, and the damage thus inflicted can result in substantial economic losses.

Climate change has attracted worldwide attention, and it is becoming important to understand the influence of the incidence and severity of climate on insect pests in agriculture [6,7,8]. Temperature is a notable abiotic factor that facilitates the behaviors, biochemical reactions, development, fecundity, and physiology of the insects and influences population dynamics and the structure and functioning of ecosystems [9,10,11,12,13,14,15]. Insects exposed to ambient temperatures above and below daily mean temperatures follow a thermal cycle. Thermal performance curves of insects show an asymmetric nature meaning the warming phase of a cycle has a more beneficial impact on the ecological and physiological responses of insects than the cooling phase [11,16,17]. Knowledge of the adaptations of insects to different climatic conditions can make an important contribution to predicting the emergence of pests.

Life table analysis can yield two types of information, namely, the essential results and derived parameters, and provides a comprehensive understanding of the life history of a population cohort. Currently the age-stage, or the two-sex life table theory, is widely used as it incorporates both sexes and accounts for variable developmental rates among individuals, as well as reveals the stage differentiation of individuals in a given population [18,19,20,21,22,23,24,25]. Numerous published studies have reported on the effects of constant temperatures on the demographic characteristics and population parameters of *R. pedestris* [10,26,27]. Kim et al. [26] investigated the temperature-dependent development and survival at seven constant temperatures and examined the longevity and reproduction of *R. pedestris* at six different constant temperatures. Although they did not show the life table parameters of *R. pedestris*, the mean total fecundity was highest at 25 °C. Ahn et al. [10] conducted an experiment on temperature-dependent development at eleven constant temperatures and for adult longevity and oviposition at six constant temperatures. Ahn et al. [10] showed that the net reproductive rate (RO) of *R. pedestris* was highest at 24 °C, although the intrinsic rate of increase was highest at 32.6 °C using methods proposed by Maia et al. [28]. However, to the best of our knowledge, there have been no studies to date that have examined the effects of fluctuating temperature conditions on the lifecycle of *R. pedestris.* Fluctuating temperatures lead to ecological, life history, and physiological consequences for insects that diverge from those predicted from constant temperatures. Fluctuating temperatures that remain within permissive temperature ranges improve biological performance [11,29,30,31,32,33,34,35]. This study hypothesized that fluctuating temperatures remaining within the permissive thermal range influence the life history parameters of *R. pedestris*. We accordingly collected detailed life history data on the development, survival, reproduction, and longevity rates of *R. pedestris* under a single constant and three fluctuating temperatures to determine the population parameters and predict the population growth of *R. pedestris* populations. Specifically, we analysed the raw life history data using age-stage, two-sex life tables and then used the data thus obtained to project population growth under different temperature conditions.

## 2. Materials and Methods

### 2.1. Insect Maintenance

A colony of *R. pedestris* was obtained from the National Institute of Agricultural Sciences (NIAS), Wanju-gun, Korea. Colonies of the bug were maintained in acrylic cages (30 cm × 30 cm × 35 cm) comprising of two meshed screens on the lateral sides for ventilation and a sliding door. The cages were maintained at 25 ± 1 °C and 60% ± 10% relative humidity under a 16 h:8 h light:dark photoperiod. Soybean seeds (var. Baegtae) and water were supplied as a food source. Three pieces of gauze (10 cm × 4.5 cm) were placed at the bottom of each cage as oviposition substrates. To prevent inbreeding depressions, field-collected *R. pedestris* from Jeju were added to cages for mass rearing.

### 2.2. Laboratory Experiment

Freshly deposited *R. pedestris* eggs (<24 h old) were randomly selected from the insect-rearing system and placed in Petri dishes (10.0 cm diameter and 4.0 cm height; SPL, Pocheon, Korea), the lids of which contained a 4.0 cm diameter air hole covered with a 0.05 mm mesh. Water-soaked cotton was placed in each dish to maintain humidity and the dishes were transferred to a controlled temperature chamber. After eclosion, the first-instar nymphs were selected with a brush and placed individually in Petri dishes (5.0 cm diameter and 1.5 cm height with a 1.3 cm-diameter air hole in the lid covered with 0.05 mm mesh). Each dish was initially lined with water-soaked cotton, and three dried soybean seeds were placed in the Petri dishes and prevented from rolling with a plastic band. Soybean seeds and water were supplied as a food source for the nymphs.

Nymphal development and survival were observed at daily intervals, with the presence of an exuvium used as evidence of molting to the next developmental stage. Sample sizes were greater than 100 eggs per treatment. The nymphs were allowed to complete development to the adult stage within the Petri dishes, with developmental time for each life stage being recorded. The dried soybean seeds were replenished whenever necessary. After the emergence of adults, male and female bugs were paired and transferred to new Petri dishes (10.0 cm diameter and 4.0 cm height) for oviposition, with rolled pieces of gauze (10 cm × 4.5 cm) lining the base of the dishes and serving as oviposition substrates. For each temperature treatment, the reproductive period, fecundity, survival, and longevity of the bugs were recorded for each individual environment until the deaths of all adults. In cases when one member of the paired bugs died earlier than its mate, the surviving individual was paired with an adult of the opposite sex derived from the colonies of mass-reared *R. pedestris.* Eggs laid by the female were removed from the containers to assess the daily fecundity and hatchability. Insects were also maintained separately in each of the controlled temperature chambers for the entire life table study period.

The developmental periods of *R. pedestris* stages from egg-to-adult emergence were assessed at temperatures of 24.0 °C, 24 ± 4 °C, 24 ± 6 °C, and 24 ± 8 °C as well as a relative humidity within the range of 65–80%. These environmental data were recorded at 1-h intervals using a HOBO data logger (Onset Computer, Co., Bourne, MA, USA). The growth chamber under constant temperature regimes held the target temperature constant (± 0.3 °C) for the entirety of the study. Rate of temperature increase or decrease varied depending on temperature treatment (Appendix A). For fluctuating temperature regimes, growth chambers were programmed with incremental temperature increases hourly depending on fluctuating temperature regimes; 24 ± 4 °C was 0.4 °C, 24 ± 6 °C was 0.8 °C, and 24 ± 8 °C was 2 °C, respectively. The incremental decrease in temperature was 1 °C at 1-h intervals under each fluctuating temperature treatment. The temperature and relative humidity within the chambers were checked per week for maintaining experimental conditions. The temperatures recorded within the cages were established to be very similar to those of the chambers (WTH-305, Daihan Scientific, Co. Ltd., Seoul, Korea) in which they were maintained.

### 2.3. Life Table Data Analysis

The life history data of *R. pedestris*, namely their developmental duration, survival, longevity, fecundity, life expectancy, and population parameters, were analyzed by implementing the age-stage, two-sex life table theory [36] and the method described by Chi [37] using the computer program TWOSEX-MSChart [38]. The age-stage-specific survival rate (*s_xj_*, where *x* = age and *j* = stage), age-specific survival rate (*l_x_*), age-stage-specific fecundity (*f_xj_*), and age-specific fecundity (*m_x_*), as well as the population parameters including net reproductive rate (*R*_o_), intrinsic rate of increase (*r*), finite rate of increase (*λ*), and mean generation time (*T*) were calculated according to Chi and Liu [36] using the following equations:$$l_{x}=\sum_{j=1}^{k}S_{xj}\;m_{x}=\frac{\sum_{j=1}^{k}S_{xj}f_{xj}}{\sum_{j=1}^{k}S_{xj}}\;R_{O}=\overset{\infty }{\underset{x=0}{\sum }}l_{x}m_{x}\;\overset{\infty }{\underset{x=0}{\sum }}e^{-r\left(x+1\right)}l_{x}m_{x}=1\;\lambda =e^{r}\;T=\frac{\ln R_{O}}{r}$$
where *k* is the number of stages. The survival rate (*s_xj_*) is defined as the probability that a newly laid egg will survive to age *x* and stage *j*, and fecundity (*f_xj_*) is the number of hatched eggs produced by a female adult at age *x.* Age-specific fecundity (*m_x_*) is calculated to take into account individuals of different stages at age *x*. The net reproductive rate (*R*_o_) is defined as the total number of offspring that an individual can produce during its lifetime, whereas the intrinsic rate of increase (*r*) is calculated using the Euler-Lotka formula with age indexed from day “0” [39]. The mean generation time (*T*) represents the period that a population requires to undergo an *R*_0_-fold increase of its original size as time approaches infinity and the population growth rate settles down to the intrinsic rate of increase. Owing to the variable development rates in many insect populations, the overlapping of stages in their life history is examined where individuals of the same age have different developmental stages [6]. The age-stage-specific life expectancy (*e_xj_*), which is the length of time an individual of age *x* and stage *j* is expected to live, was calculated as follows:$$e_{xj}=\sum_{i=x}^{\infty }\sum_{y=j}^{k}s\prime_{iy}$$
where $e_{xj}$ is the probability that an individual of age *x* and stage *j* will survive to age *i* and stage *y,* which is calculated by assuming $s\prime_{iy}$ = 1.

The age-stage reproductive value (*v_xj_*) is the contribution of an individual of age *x* and stage *j* to the future population, and is calculated as follows [40]:$$v_{xj}=\frac{e^{r\left(x+1\right)}}{s_{xj}}\;\overset{\infty }{\underset{i=x}{\sum }}e^{-r\left(i+1\right)}\overset{k}{\underset{y=j}{\sum }}s\prime_{iy}f_{iy}$$

The adult pre-oviposition period (APOP) is defined as the pre-oviposition period based on adult female age, whereas the total pre-oviposition period (TPOP) takes into consideration the total time from birth to the initial oviposition. The standard errors of development time, longevity, fecundity, and population parameters were calculated using the bootstrap method with 100,000 bootstrap replicates. Differences among the different treatments were analyzed using a paired bootstrap test of the TWOSEX-MSChart program at the 5% significance level [20,41].

### 2.4. Population Projection

To predict and compare the population growth and age-stage structure of *R. pedestris* reared under constant and fluctuating temperatures, we used life table data for the developmental periods, survival rate, and fecundity to simulate population growth in the TIMING-MSChart program [42]. For comparative purposes, the same initial population of 10 newly laid eggs was used for the simulation of each treatment. The data file for TIMING-MSChart was created using TWOSEX-MSChart to reduce the complex process of preparing data files for TIMING-MSChart. We projected the increase rate of stage *j* from time *t* to *t* + 1 using the common logarithm, as the stage differentiation during population growth is described by the TWOSEX-MSChart analysis. We calculated the rate of increase of stage *j* from time *t* to *t* + 1 by using a natural logarithm [8]:$$\varphi_{j,\;t}=\ln\left(\frac{n_{j,\;t+1}+1}{n_{j,\;t}+1}\right)=\ln\left(n_{j,\;t+1}+1\right)-\ln\left(n_{j,\;t}+1\right)$$

Given that it is not possible to use logarithmic transformation when the number of individuals at a certain stage is 0 (*n_j,t_* = 0 or *n_j,t+1_* = 0), we used *n_j,t_* +1 and *n_j,t + 1_* +1 in the calculation process.

## 3. Results

Parameters related to the development of each life stage and the reproduction of the adult females under the four different controlled temperature conditions are summarized in Table 1. The pre-adult duration ranged from 34.0 d at 24 ± 6 °C to 41.4 d at 24 ± 4 °C, whereas the longevity and oviposition periods of the adult females increased with increasing temperature fluctuation until 24 ± 6 °C. The shortest reproductive period (35.4 d) and the APOP (5.3 d) were observed at 24 °C, and the highest mean fecundity was recorded at 24 ± 6 °C (321.1 eggs/female), followed by that at 24 °C (221.6 eggs/female), 24 ± 4 °C (205.2 eggs/female), and 24 ± 8 °C (195.2 eggs/female). The probability that a newly laid egg would survive to the adult stage was 0.83, 0.60, 0.53, and 0.52 at 24 ± 6 °C, 24 °C, 24 ± 8 °C, and 24 ± 4 °C, respectively, and these values were found to coincide with respective proportions of reproductive adult female individuals (*N_f_*/*N*) in the cohort (Table 2).

The age-stage-specific survival rate (*s_xj_*) is the probability that a newborn bean bug will survive to age *x* and stage *j* (Figure 1). The curves of each life stage show the survivorship, stage differentiation, and overlap among stages due to the variable rates of development among individuals. Adult females emerged at 32, 36, 31, and 33 d and survived until 136, 186, 213, and 196 d at 24.0 °C, 24 ± 4 °C, 24 ± 6 °C, and 24 ± 8 °C, respectively.

The age-specific survival rate and fecundity of *R. pedestris* are presented in Figure 2 and Figure 3, respectively. The age-specific survival rate (*l_x_*) is the sum of *s_xj_* at each age *x* and is thus a simplified version of *s_xj_* shown in Figure 1. As shown in Figure 2, with exception of those individuals exposed to 24 ± 6 °C, the curves decline sharply at a relatively early age. The fecundity curve, *m_x_*, ended at ages 113, 164, 165, and 160 d at 24.0 °C, 24 ± 4 °C, 24 ± 6 °C, and 24 ± 8 °C, respectively. The period from the first to the final reproduction under the four different temperature treatments ranged from 78 d (24.0 °C) to 132 d (24 ± 6 °C), whereas the highest age-specific fecundities were 4.11 (45 d), 2.54 (66 d), 3.86 (46 d), and 3.08 (47 d) eggs at 24.0 °C, 24 ± 4 °C, 24 ± 6 °C, and 24 ± 8 °C, respectively.

The life expectancy (*e_xj_*) of *R. pedestris* at different ages and stages under the four different temperature treatments is plotted in Figure 4. Each plot represents the expected survival of an individual at age *x* and stage *j*. The curves for females and males show a decline with age. The life expectancy of females was 42.4, 67.9, 94.9, and 82.1 d at 24.0 °C, 24 ± 4 °C, 24 ± 6 °C, and 24 ± 8 °C, respectively.

The age-stage-specific reproductive value (*v_xj_*) represents the contribution of an individual bean bug of age *x* and stage *j* to the future population (Figure 5). This value significantly increased with an increase in female emergence, and it peaked when females began to produce eggs. Major peaks in the reproductive values of the females maintained at 24.0 °C, 24 ± 4 °C, 24 ± 6 °C, and 24 ± 8 °C were observed at 49 (80.03), 60 (71.60), 43 (67.69), and 42 d (61.58), respectively, and accordingly, the females at these ages contributed the most to the next generation.

The derived population parameters assessed in this study are presented in Table 2. The highest net reproductive rate (*R*_0_; 159.1 offspring) was observed at 24 ± 6 °C, whereas the lowest *R_0_* value (53.2 offspring) was obtained for females reared at 24 ± 4 °C. The highest intrinsic (*r* = 0.09 d^−1^) and finite (*λ* = 1.09 d^−1^) rates of increase for *R. pedestris* were recorded at 24 ± 6 °C, whereas the longest mean generation time (*T* = 69.8 d) was recorded at 24 ± 4 °C, which was 14 days longer than the *T* value (55.7 d) at 24 °C.

The population growth and stage structure at different temperature treatments based on age-stage, two-sex life table theory are shown in Figure 6. The populations subjected to the 24 ± 6 °C treatment were found to grow more rapidly than those exposed to the other three temperature conditions. The curves of the stage-specific growth rates of *R. pedestris* subjected to the different temperature treatments approached the intrinsic rate of increase for each temperature condition (0.08, 0.05, 0.09, and 0.06 for 24.0 °C, 24 ± 4 °C, 24 ± 6 °C, and 24 ± 8 °C, respectively; Table 2 and Figure 7).

## 4. Discussion

The primary objective of this study was to determine the impact of fluctuating temperatures on the development and reproduction of the bean bug *R. pedestris*. The effects of thermal conditions were assessed by constructing age-stage, two-sex life tables for sample populations exposed to a constant temperature of 24 °C and simulated fluctuating temperatures of 24 ± 4 °C, 24 ± 6 °C, and 24 ± 8 °C. Our results showed that some life table parameters of *R. pedestris* were different from fluctuating temperature treatments compared to the constant temperature treatment. Compared to the constant temperature treatment, pre-adult development of *R. pedestris* was similar, but female longevity and oviposition period were significantly longer in fluctuating temperature treatments. These findings are partially consistent with the results of earlier studies on *Bicyclus anynana* [43], *Helicoverpa armigera* [44], *Tetranychus urticae*, *Phytoseiulus persimilis*, *Neoseiulus californicus* [45], and *Megoura crassicauda* and *Aphis craccivora* [46].

A number of studies have previously highlighted that experiments focusing primarily on the influence of average temperatures may not be sufficiently comprehensive to explain the variability of fluctuating temperature influences on insect thermal fitness [29,30,31,32,33,34,35,47,48,49,50]. Consequently, it is necessary to determine how fluctuations in temperature influence the thermal fitness of insects to gain insight into their biological responses when exposed to changing environments [49,51,52,53]. Cheng et al. [46] examined the impact of fluctuating temperatures on the development and reproduction of *Megoura crassicauda* and *Aphis craccivora*. They accordingly established that fluctuating temperatures accelerated the development of *M. crassicauda* and *A. craccivora* by significantly shortening the preadult period compared to the corresponding constant temperature. Furthermore, these authors found that fluctuating temperatures (22 ± 3 °C and 22 ± 5 °C) reduced the rate of adult *M. crassicauda* survival but increased that of *A. craccivora* [46]. Similarly, whereas reductions were observed in the oviposition period and fecundity of *M. crassicauda* under both fluctuating temperature treatments, increases were recorded for *A. craccivora* under the same conditions. Although no significant differences were detected between the species with respect to the population growth rate (intrinsic rate of increase) under the constant temperature, the simulated *A. craccivora* population grew significantly more rapidly than the *M. crassicauda* population when subjected to the two fluctuating temperature treatments [46]. Collectively, these findings indicated that fluctuating temperatures have detrimental effects on the life history traits of *M. crassicauda*, whereas the same conditions appear to be beneficial for *A. craccivora*. Another similar study by Bahar et al. [54] reported that the developmental times of the diamondback moth *Plutella xylostella* (Linnaeus) and its larval parasitoid *Diadegma insulare* (Cresson) were shortened under fluctuating temperatures (0–14, 15–29, and 23–37 °C) compared with those under the corresponding constant temperatures of 7, 22, and 30 °C, respectively. These findings may indicate that the effects of fluctuating temperatures on the population growth rate could be species-specific. Similarly, fluctuating temperatures appear to enhance the population growth rates of *Spodoptera frugiperda* [55], *Helicoverpa armigera* (Hübner) [44], *Myzus persicae* [47], and *Tetranychus urticae* Koch [45] and adversely affect *Anagasta kuehniella* [56]. Fluctuating temperatures may alter fitness components, including fecundity and longevity, dependent on insect sensitivity [11,57].

The intrinsic rate of increase includes contributions of age of initial reproduction, the peak of reproduction, length of the reproductive period, and the survival rate of the population. Population parameters are principal demographic parameters and are particularly useful when predicting the growth potential of insect populations [45,58,59,60,61,62,63]. Based on the intrinsic rate of increase values obtained in the present study, we established that fluctuating temperatures have different effects on *R. pedestris* development and reproduction, dependent on the ranges of temperature fluctuation to which these bugs were exposed. The higher values obtained in response to the 24 ± 6 °C can plausibly be attributed to a shorter period of immature development and higher adult fecundity. Cheng et al. [46] reported that fluctuating temperatures had detrimental effects on the development and fertility of *M. crassicauda*, but they were beneficial for *A. craccivora*. Fluctuating temperatures have also been found to increase the population growth rates of *H. armigera* (Hübner) [44] and *T. urticae* Koch [45], whereas Rismayani et al. [64] observed a higher intrinsic rate of increase in *Tetranychus pacificus* under fluctuating temperatures around 10, 20, and 30 °C than under the corresponding constant temperatures; the converse was true in the case of *T. pacificus* exposed to constant temperatures of 25 and 35 °C.

Collectively, our findings indicate that the developmental periods, survival rates, longevity, reproductive capacities, and population growth of *R. pedestris* under fluctuating temperatures can often differ from those observed in response to a constant temperature. Fluctuating temperatures may create more resource-consuming environments than constant temperatures held at equivalent mean temperatures and tend to produce divergent responses. Given that temperature is one of the most important factors used in predicting population dynamics in nature, the effects of fluctuating temperatures should be carefully taken into account when attempting to forecast the population growth of *R. pedestris*.

## 5. Conclusions

In this study, we obtained fundamental information pertaining to the thermal development of all the life stages, life table parameters, and the population growth of *R. pedestris* reared under constant and fluctuating temperature conditions. The effects of thermal conditions were assessed by constructing age-stage, two-sex life tables for populations exposed to a constant temperature of 24 °C and simulated fluctuating temperatures of 24 ± 4 °C, 24 ± 6 °C, and 24 ± 8 °C. We accordingly established that values obtained for a number of life table parameters of *R. pedestris* under constant temperature conditions differed from those obtained for populations exposed to fluctuating temperatures. Based on our findings, we believe that determining the population parameters and growth of *R. pedestris* under both constant and fluctuating temperature conditions would make a valuable contribution to predicting population fluctuations in field populations of *R. pedestris* and thus provide important information for implementing appropriate management practices. There are other factors such as photoperiod, diet, relative humidity, rearing density, sex, and all the interactions between these factors influencing on development rate in insects. Future studies may be needed to examine the biological performances of *R. pedestris* under different environmental conditions affecting insect development and fecundity.

## Figures and Tables

**Figure 1 insects-13-00113-f001:**
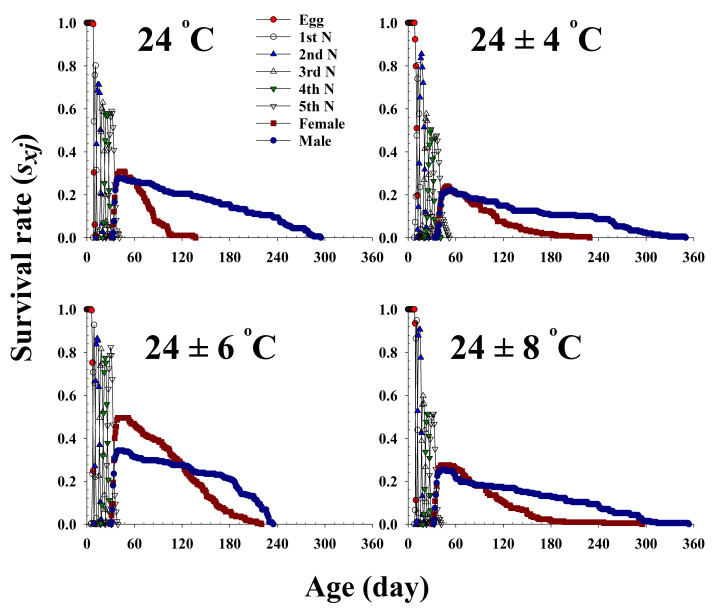
The age-stage-specific survival rate (s*_xj_*) of *Riptortus pedestris* in response to different temperature conditions.

**Figure 2 insects-13-00113-f002:**
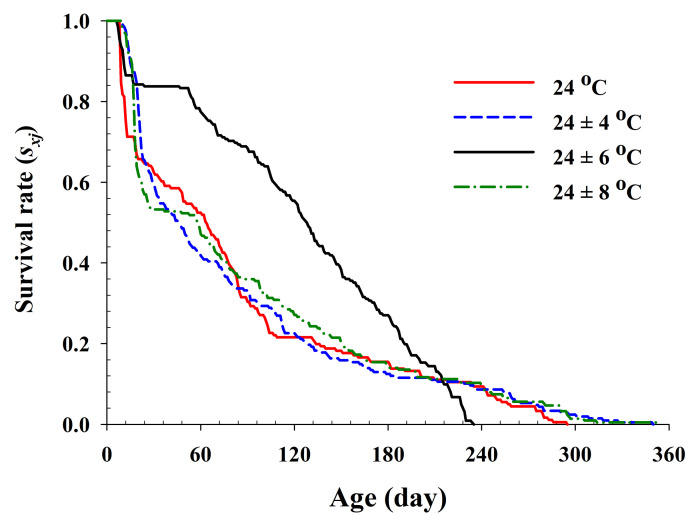
The age-specific survival rate (*l_x_*) of *Riptortus pedestris* in response to different temperature conditions.

**Figure 3 insects-13-00113-f003:**
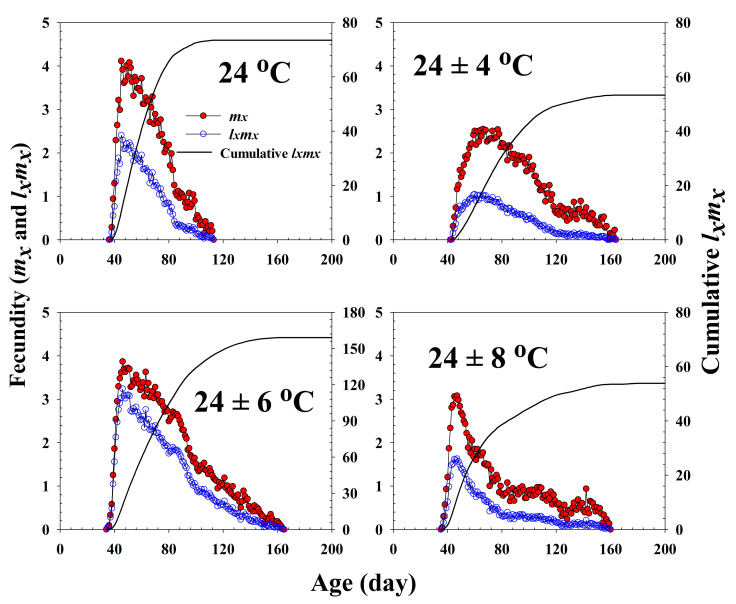
The age-specific fecundity (*m_x_*), the age-specific maternity (*l_x_m_x_*), and the cumulative reproductive rate ($R_{x}=\sum l_{x}m_{x}$) of *Riptortus pedestris* in response to different temperature conditions.

**Figure 4 insects-13-00113-f004:**
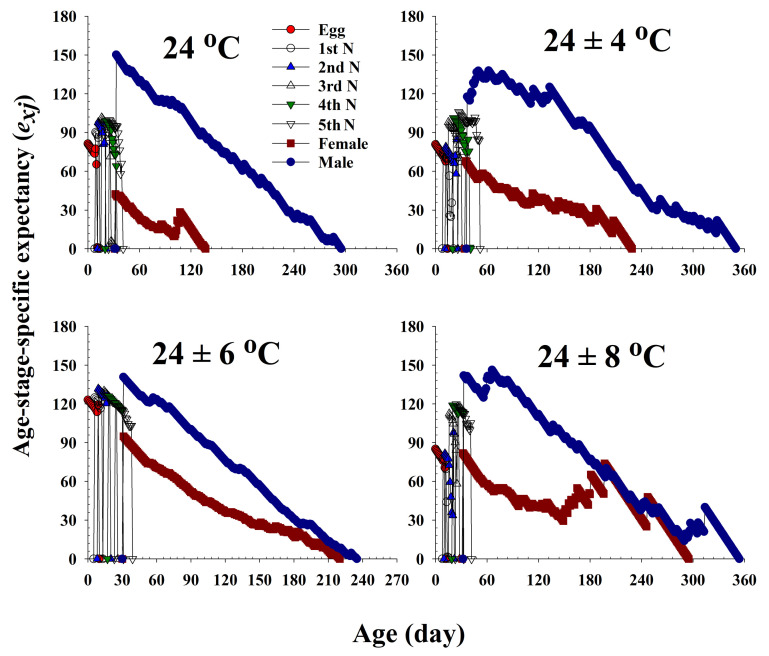
The age-stage life expectancy (*e_xj_*) of *Riptortus pedestris* in response to different temperature conditions.

**Figure 5 insects-13-00113-f005:**
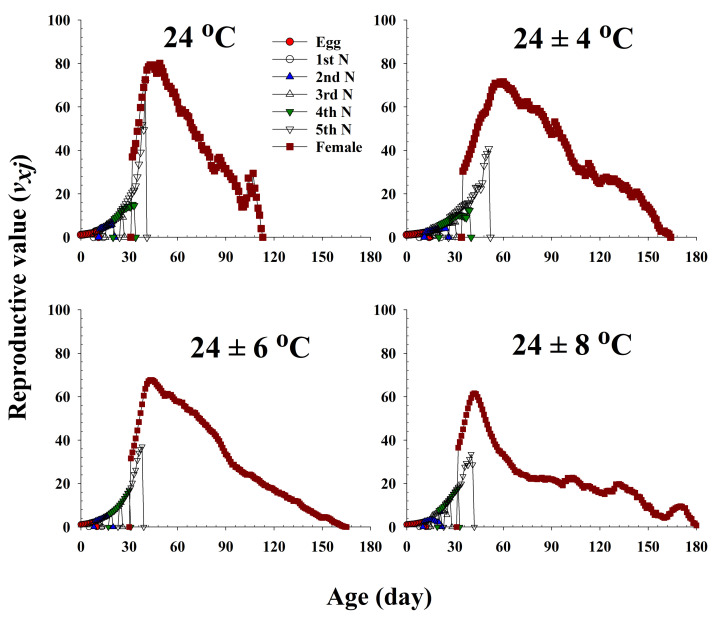
The age-stage-specific reproductive value (*v_xj_*) of *Riptortus pedestris* in response to different temperature conditions.

**Figure 6 insects-13-00113-f006:**
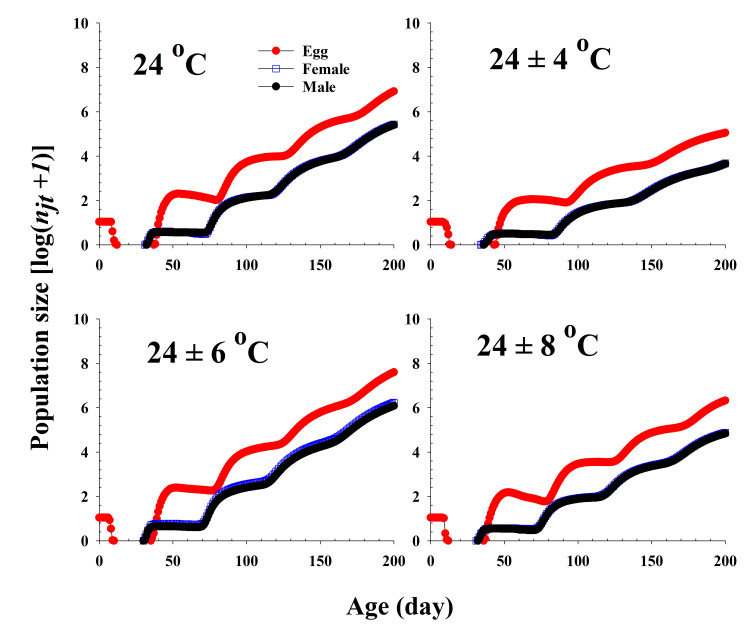
Projection of *Riptortus pedestris* population growth in response to different temperature conditions commencing with an initial population of newly laid 10 eggs.

**Figure 7 insects-13-00113-f007:**
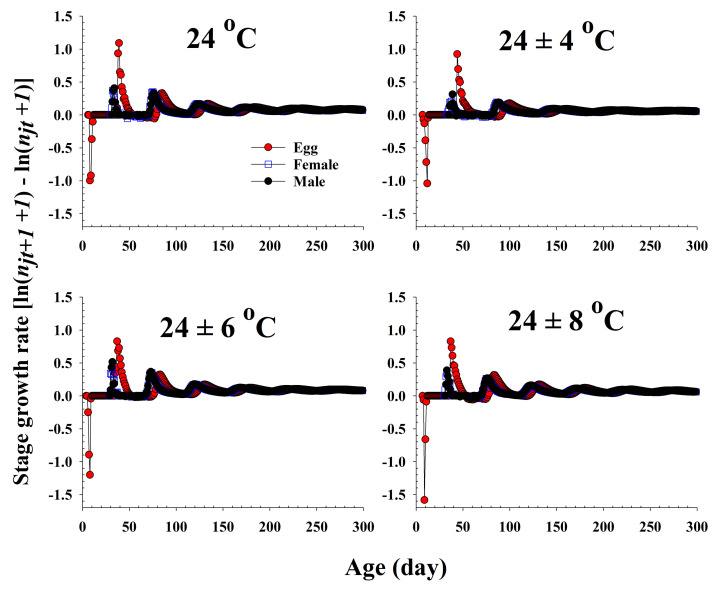
The stage growth rate of *Riptortus pedestris* in response to different temperature conditions.

**Table 1 insects-13-00113-t001:** Development time, longevity, adult pre-oviposition period (APOP), total pre-oviposition period (TPOP), oviposition days, eggs per oviposition day, and fecundity of all females of *Riptortus pedestris* under four different temperature conditions.

Parameters	Temperatures (°C)			
	24	24 ± 4	24 ± 6	24 ± 8
Egg P	9.4 ± 0.04c (145)	11.4 ± 0.08a (201)	8.0 ± 0.05d (208)	10.1 ± 0.03b (209)
1st instar P	3.0 ± 0.02b (129)	3.2 ± 0.04a (181)	2.9 ± 0.03c (192)	2.4 ± 0.03d (194)
2nd instar P	5.6 ± 0.09b (119)	6.1 ± 0.07a (136)	5.3 ± 0.003c (187)	4.9 ± 0.07d (131)
3rd instar P	5.0 ± 0.07b (116)	5.5 ± 0.09a (124)	4.8 ± 0.05b (187)	5.0 ± 0.07b (119)
4th instar P	5.2 ± 0.06c (110)	6.1 ± 0.10a (110)	5.3 ± 0.002bc (186)	5.5 ± 0.06b (114)
5th instar P	7.1 ± 0.07d (110)	9.1 ± 0.20a (110)	7.5 ± 0.04c (186)	7.8 ± 0.06b (114)
Pre-adult P	35.4 ± 0.17b (110)	41.4 ± 0.35a (110)	34.0 ± 0.13c (186)	35.6 ± 0.18b (114)
Adult female P	40.7 ± 2.74c (60)	62.0 ± 5.76b (54)	91.8 ± 4.05a (110)	78.5 ± 6.14a (59)
Adult male P	147.8 ± 10.50a (50)	120.5 ± 13.26a (56)	137.9 ± 6.62a (76)	140.2 ± 12.42a (55)
APOP	5.3 ± 0.11b (56)	9.9 ± 0.71a (54)	5.3 ± 0.09b (109)	5.7 ± 0.78b (59)
TPOP	40.8 ± 0.29b (56)	51.1 ± 0.92a (54)	39.5 ± 0.19c (109)	41.2 ± 0.82b (59)
Oviposition P	35.4 ± 2.24c (60)	41.5 ± 4.42bc (54)	69.6 ± 2.97a (110)	44.9 ± 2.49b (59)
Fecundity (eggs/female)	221.6 ± 15.88b (60)	205.2 ± 23.54b (54)	321.1 ± 11.09a (110)	195.2 ± 9.76b (59)

P: period. Means in the same row followed by the same letter do not differ significantly at the *p* < 0.05 level, as determined using the paired bootstrap test.

**Table 2 insects-13-00113-t002:** Population parameters and proportions of females, males, and N-type individuals of *Riptortus pedestris* under four different temperature conditions.

Parameters	Temperatures (°C)			
	24	24 ± 4	24 ± 6	24 ± 8
Pre-adult survival rate (%)	60.8 ± 0.03b (181)	52.8 ± 0.03b (208)	83.8 ± 0.02a (222)	53.2 ± 0.03b (214)
First age survival rate <50%	63.1 ± 7.07b (181)	45.3 ± 6.34b (208)	127.7 ± 5.31a (222)	52.4 ± 13.75b (214)
Net reproductive rate (*R_o_*)	73.4 ± 9.37b (181)	53.2 ± 8.67b (208)	159.1 ± 12.07a (222)	53.8 ± 6.53b (214)
Intrinsic rate of increase (r)	0.08 ± 0.002b (181)	0.05 ± 0.002d (208)	0.09 ± 0.001a (222)	0.06 ± 0.002c (214)
Finite rate of increase (λ)	1.08 ± 0.002b (181)	1.05 ± 0.002d (208)	1.09 ± 0.001a (222)	1.07 ± 0.002c (214)
Mean generation time (T)	55.7 ± 0.63c (181)	69.8 ± 1.14a (208)	59.1 ± 0.43b (222)	57.2 ± 0.91bc (214)
Proportion of female individuals (N*_f_*/N)	33.2 ± 0.03b (181)	26.0 ± 0.03b (208)	49.6 ± 0.03a (222)	27.6 ± 0.03b (214)
Proportion of male individuals (N*_m_*/N)	27.6 ± 0.03ab (181)	27.0 ± 0.03ab (208)	34.2 ± 0.03a (222)	25.7 ± 0.02b (214)
Proportion of N-type individuals (N*_n_*/N)	39.2 ± 0.03a (181)	47.1 ± 0.03a (208)	16.2 ± 0.03b (222)	46.7 ± 0.03a (214)

Means in the same row followed by the same letter do not differ significantly at the (*p* < 0.05) level, as determined by the paired bootstrap test. N-type means *R. pedestris* could not develop to adult stage.

## Data Availability

Data presented in this study are available in the article.

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
