# Peer review of "Population Parameters and Growth of Riptortus pedestris (Fabricius) (Hemiptera: Alydidae) under Fluctuating Temperature"

_insects, 2022, doi:10.3390/insects13020113_

Round 1
Reviewer 1 Report
Dear,
The manuscript entitled ‘Population parameters and growth of Riptortus pedestris (Fabricius) (Hemiptera: Alydidae) under fluctuating temperature’ has been reviewed. It has an interesting subject, good writing, sound experimental methods, and the presentation of the results is well done. However, it seems necessary to make the following corrections before the final acceptance of the article.
Comments for revision:
Line 9: All Asian countries or some of them? It has no economic importance in Western Asia.
Lines 51-53: Numerous published studies have reported on the effects of constant temperatures ... . Add more details about constant temperatures and their effects on population parameters and life history of the insect pest (at least two examples). It will be a good background. The INTRODUCTION is weak and must be modified. Susceptibility of Riptortus pedestris to different temperatures and even other environmental conditions should also be considered.
Line 69: Which cultivar of soybean was used?
The conclusion is general and optimistic. Is forecasting based on temperature only? Other factors, such as humidity, are certainly important. Mention the possible impact of other factors in the explanation of the conclusion.
I think the references are not written according to the journal's instructions and recently published papers in INSECTS. Please check them.
Best regards
Author Response
Response to Reviewers’ Comments
All authors really appreciate two anonymous reviewers for valuable comments on our manuscript.
Response to Reviewer 1’s Comments
The manuscript entitled …the article:
[RE] The authors appreciate your positive review. We revised some parts of manuscript as suggested.
Comments for revision
Line 9: All Asian countries …in Western Asia.
[RE] We thank for pointing it out. We revised it as suggested. Please refer to line 39-40.
Line 51-53: Numerous published studies….also be considered
[RE] We thank for pointing it out. We revised it as suggested. Please refer to line 53-61.
Line 69: Which cultivar of soybean was used?
[RE] We thank for pointing it out. We added the name of cultivar. Please refer to line 77.
The conclusion is general and optimistic…the conclusion.
[RE] We respect the comment. We revised some contents of conclusion as suggested. Please refer to line 375-379.
I think the references are not written …please check them.
[RE] We thank for pointing it out. We revised and added the references as suggested. Please refer to line 498-505.

Reviewer 2 Report
These are my main comments on the MS (insects-1538120) entitled: "Population parameters and growth of Riptortus pedestris (Fabricius) (Hemiptera: Alydidae) under fluctuating temperature" by Jeong Joon Ahn and Kyung San Choi.
The introduction and discussion provide no insight on how this MS relates to the various other ones cited in the text or concerns that have been raised by other researchers. This article should provide details on all these fronts to provide the proper context for the work. Authors do not present any hypotheses or expectations that could be connected to previous studies; adding these details will improve the paper. The authors should clearly explain WHY THE STUDY WAS DONE, WHY IT WAS IMPORTANT, and HOW IT FITS WITH OTHER STUDIES INVESTIGATING THE EFFECTS OF CONSTANT AND FLUCTUTAING TEMPRATURES ON INSECT DEVELOPMENT, LONGEVITY, etc., etc. It should be clear and concise. The discussion should also include what outcome(s) they expect, and how it would help support or refute their hypotheses or answer their questions.
My major concern is that the authors are extrapolating the applicability of their results beyond what the design supports. These are only data from a narrow set of highly artificial laboratory conditions, so the inference power of the paper is very limited, but authors do not acknowledge this detail at all and need to be more forthcoming. Only a single constant temperature regime of 24C was investigated, and the data were compared to those obtained at three sets of highly artificial fluctuating temperature regimes programed to ramp in increments within 24±4C, ±6C, and ±8C. The development and life history parameters of R. pedestris at temperature profiles (constant and fluctuating) lower and higher then 24C were not investigated in this study. This is a critical limitation of the study, and the authors must concede and discuss this. It creates major bias in your validation analysis and future recommendations. The interaction of cyclic temperatures with nonlinear development or life history parameters can introduce significant deviations from the results obtained here, especially at the lower and higher temperatures of the development rate, viability, and reproductive activity functions. The consequences of getting this wrong will affect real people and livelihoods. Therefore, studies across a broader set of fluctuating temperature regimes are still necessary to understand the real effect of temperature on the characteristics of this pest, as this is the closest to the daily temperature fluctuations that occur in the field. So, I am suggesting to the authors to tone-down the language a little and admit that there are still substantive uncertainties to be considered, including uncertainty as to how generalizable the results are to open field conditions.
Some of the authors statements would be much stronger if they tie their work to the body of literature that has built up from rearing insects at temperature regimes that fluctuate over 24h cycles (see examples from J. Econ. Entomol. 2019, 112: 1560-1574; J. Econ. Entomol. 2019, 112:1062-1072). This is not to diminish the data gathered in this study, they are of value. But it is important for the authors not to overgeneralize, and to warn the reader, including regulatory agencies, against doing so as well. Adding these details will improve the paper.
Overall, I was excited to see the results of the paper after reading the abstract, but I found it hard to extract key messages useful to policymakers and professionals, probably in large part due to the lack of connection with other published work and need for improved structure of the current manuscript.
The next draft of this paper will need to be dramatically different to have a chance at publication in my opinion.
Additional specific comments:
L99-102: Detailed methods for fluctuating temperature profile selections/programming are also missing. Were incremental steps based on the hourly or XX temperature profiles over 24h. Please explain.
L100: Speaking from personal experience, climate-controlled cabinets tend to oscillate around the desired/prescribed mean temperature at least ±2C, especially if used for a longer period, such is the case here. How was the target temperature confirmed over time? … with HOBO loggers placed within temperature cabinets, which automatically measured temperature XX min intervals for the duration of experiments? Also, how was the oscillation in RH over time controlled? Please explain.
Author Response
Response to Reviewers’ Comments
All authors really appreciate two anonymous reviewers for valuable comments on our manuscript.
Response to Reviewer 2’s Comments
These are …open field conditions.
[RE] The authors appreciate your positive review. We revised some parts of manuscript as suggested and added supplementary figure. Please refer to line 53-61, 109-120, 375-379 and supplementary materials.
Additional specific comments
Line 99-102: Detailed methods for fluctuating temperature profiles….explain.
[RE] We thank for pointing it out. We revised it as suggested. Please refer to line 109-120.
Line 100: Speaking from…explain.
[RE] We thank for pointing it out. We revised it as suggested. Please refer to line 109-120 and supplementary figure.
Round 2
Reviewer 1 Report
Dear
I have no more comments. I think the manuscript is acceptable.
Please consider only the following:
In the newly added sentences, please italicize the R. pedestris in line 60.
Best regards
Author Response
Response to Reviewers’ Comments
All authors really appreciate two anonymous reviewers for valuable comments on our manuscript.
Response to Reviewer 1’s Comments
I have no more comments. ….acceptable.
[RE] The authors appreciate your positive comments.
In the newly added, in line 60.
[RE] We thank for pointing it out. We revised it as suggested. Please refer to line 67.

Reviewer 2 Report
The authors have failed to address most of my major comments and concerns, if not all, about this manuscript and I wonder why this is the case.
The introduction and discussion still do not provide any insight on how this MS relates to the various other ones cited in the text or concerns that have been raised by other researchers. The authors do not present any hypotheses or expectations that could be connected to previous studies; adding these details will improve the paper. The authors should clearly explain WHY THE STUDY WAS DONE, WHY IT WAS IMPORTANT, and HOW IT FITS WITH OTHER STUDIES INVESTIGATING THE EFFECTS OF CONSTANT AND FLUCTUTAING TEMPRATURES ON INSECT DEVELOPMENT, LONGEVITY, etc., etc. It should be clear and concise. The discussion should also include what outcome(s) they expect, and how it would help support or refute their hypotheses or answer their questions. This was not addressed in the revised manuscript, and I wonder why.
The methodology has been improved and better explained, thank you!
The inference power of the paper is very limited, because of the narrow set of temperatures examined, but authors do not acknowledge this detail at all and need to be more forthcoming. Only a single constant temperature regime of 24C was investigated, and the data were compared to those obtained at three sets of highly artificial fluctuating temperature regimes programed to ramp in increments within 24±4C (i.e., 20-28C), ±6C (i.e., 18-30C), and ±8C (i.e., 16-32C). The 24-32C range is within the optimal range for development and fitness outcomes of this species according to previous studies. The effects of constant and fluctuating temperatures, outside of this optimum, and closer to the lower (<16C) and upper (>32C) thermal thresholds of this species were not investigated in this study. This is a critical limitation of the study, and the authors must concede and discuss this. It creates major bias in your validation analysis and future recommendations. The interaction of cyclic temperatures with nonlinear development or life history parameters can introduce significant deviations from the results obtained here, especially at the lower (<16C) and higher (>32C) temperatures of the development rate, viability, and reproductive activity functions. The consequences of getting this wrong will affect real people and livelihoods. Therefore, studies across a broader set of fluctuating temperature regimes are still necessary to understand the real effect of temperature on the characteristics of this pest, as this is the closest to the daily temperature fluctuations that occur in the field (and not only ramping between 24-32C). I am still suggesting to the authors to tone-down the language a little and admit that there are still substantive uncertainties to be considered, including uncertainty as to how generalizable the results are to open field conditions.
Like I mentioned in my original review, most of the authors statements would be much stronger if they tie their work to the body of literature that has built up from rearing insects at temperature regimes that fluctuate over 24h cycles like in this study. Please read the suggested examples from J. Econ. Entomol. 2019, 112: 1560-1574; and J. Econ. Entomol. 2019, 112:1062-1072). Adding these details will improve the paper.
I still found it hard to extract key messages useful to policymakers and professionals, probably in large part due to the lack of connection with other published work and need for improved structure of the current manuscript.
The next draft of this paper will need to be dramatically different to have a chance at publication in my opinion.
Author Response
Response to Reviewers’ Comments
All authors really appreciate two anonymous reviewers for valuable comments on our manuscript.
Response to Reviewer 2’s Comments
The authors …. why this is the case.
[RE] The authors appreciate your positive review. We revised some parts of manuscript as suggested. Please refer to line 43-51, 61-68, 71-76, 352-354, 375-377, and 412-416.
Like I mentioned .. the paper.
[RE] The authors respect your comments. We answered the reviewer’s question with sincerely. We cited two papers, which the reviewer2 recommended to read for improving the manuscript, and those papers help to revise the manuscript.
